# Simulated sample splitting approach to address biases due to instrument selection and participant overlap in two-sample Mendelian Randomization studies

Amanda Forde[ID][1,☯*], Gibran Hemani[ID][2,3☯], John Ferguson[1☯]

**1** School of Mathematical and Statistical Sciences, University of Galway, Galway, Ireland, **2** NIHR Bristol Biomedical Research Centre, University Hospitals Bristol and Weston NHS Foundation Trust and University of Bristol, Bristol, United Kingdom, **3** MRC Integrative Epidemiology Unit (IEU), Bristol Medical School, University of Bristol, Bristol, United Kingdom

☯ These authors are contributed equally to this work.
* amanda.forde@universityofgalway.ie

## Abstract

Mendelian randomization (MR) is a popular statistical technique that uses genetic variants to explore causal relationships in observational epidemiology. Summary-level MR, the most common form, relies on published GWAS summary statistics to estimate causal effects between exposures and outcomes. However, empirical analyses tend to ignore issues relating to Winner's Curse of instrument effects, weak instrument bias and sample overlap. Our simulations and empirical analyses using the UK Biobank indicate that such mechanisms can induce substantial bias in routine MR approaches. We propose MR Simulated Sample Splitting (MR-SimSS), a novel method that corrects this bias requiring no additional data beyond GWAS summary statistics for the exposure and outcome of interest. It operates by simulating statistically independent sets of summary statistics, analogous to what would be produced by splitting the individual-level data into independent subsets, which can then be plugged into existing two-sample MR methods. With sufficient instrument variants, MR-SimSS is robust to a range of sample overlap scenarios, providing a practical and modular solution to Winner's Curse and weak instrument bias.

## Author summary

A central challenge in epidemiology is determining whether an observed association reflects a true cause-and-effect relationship. Mendelian randomization (MR) addresses this by using genetic variants as natural experiments to test whether a particular trait or exposure genuinely influences disease risk. However, when the same genetic data are used both to select and to estimate genetic instruments, MR results can become biased due to a phenomenon known as the

**Data availability statement:** The code used for both the simulation study and the real-data analysis in the manuscript is available at https://github.com/amandaforde/mrsimss-paper. Code to implement MR-SimSS is available in the 'mr.simss' R package (https://github.com/amanda-forde/mr.simss). The real-data analysis has been conducted using the UK Biobank Resource under Application Number 23739 (https://www.ukbiobank.ac.uk/enable-your-research/approved-research/exploring-the-shared-genetic-aetiology-between-schizophrenia-and-cognition).

**Funding:** AF is funded by Science Foundation Ireland (https://www.sfi.ie/) under award 18/CRT/6214. The funders had no role in study design, data collection and analysis, decision to publish, or preparation of the manuscript.

**Competing interests:** The authors have declared that no competing interests exist.

Winner's Curse. This problem, along with weak instruments and sample overlap between datasets, can distort causal estimates even in large studies. We introduce MR Simulated Sample Splitting (MR-SimSS), a new framework that overcomes these issues using only publicly available genome-wide association study (GWAS) summary statistics. MR-SimSS works by statistically simulating independent subsets of the data, without requiring access to individual-level information, allowing existing MR methods to be applied without bias. Through extensive simulations and analyses using UK Biobank data, we show that MR-SimSS provides more accurate and reliable causal estimates, offering a practical tool for robust causal inference in modern genetic epidemiology.

## Introduction

Mendelian randomization (MR) is a statistical framework that uses genetic variation to assess whether a modifiable exposure causally influences an outcome of interest, and to estimate the magnitude of this effect [1]. Since its advent, the use of MR in epidemiological research has grown exponentially, largely due to the limitations of traditional observational studies, such as unmeasured confounding and reverse causation [2]. MR capitalises on the principle that genetic variants are fixed at conception and randomly inherited, rendering MR analyses approximately analogous to 'naturally occurring randomized trials' [3]. It relies on instrumental variable (IV) estimation, where genetic variants serve as instruments. For a variant to be a valid IV, it must be associated with the exposure, independent of confounders and influence the outcome only through the exposure pathway [4].

With the widespread availability of large-scale GWAS summary data, most MR analyses now use a summary-level design. This involves performing the MR analysis using publicly available estimates of variant-exposure and variant-outcome associations, along with their standard errors, typically derived from two non-overlapping, or partially overlapping, GWAS datasets [5]. Provided certain conditions hold and a valid genetic instrument is used, a consistent estimate of the causal effect can be obtained by dividing the variant-outcome association by the variant-exposure association [6]. To increase statistical power, ratio estimates across multiple genetic variants are often aggregated using meta-analytic methods such as the inverse variance weighted (IVW) estimator [7]. However, IVW is known to be sensitive to weak instruments and violations of the IV assumptions [8]. To address these issues, alternative summary-level methods, such as MR-Egger [9], MR weighted median [6] and MR-RAPS [10], have been proposed.

A key focus of this work is the mitigation of Winner's Curse bias in summary-level MR. To satisfy the IV relevance condition, genetic variants are usually selected by applying a genome-wide significance threshold, e.g., p-value $< 5 \times 10^{-8}$, to exposure GWAS summary data. However, using the same dataset for both instrument selection and estimation introduces Winner's Curse, where variant-exposure associations of selected variants are overestimated due to selection bias [11]. Provided that the variant-outcome associations are estimated independently, this results in downward

bias in the MR causal effect estimate, pulling it toward the null. To date, the only widely adopted solution to this problem has been the use of a third, independent dataset for instrument selection in a so-called 'three-sample' design [10]. While this avoids the overlap that causes Winner's Curse, it is often impractical, requiring access to three large, non-overlapping GWAS datasets of similar ancestry, which is a challenge for many traits, particularly in understudied populations. Alternatively, a single dataset could be split into separate parts, but this sacrifices statistical power and efficiency, and also is only an option if individual-level data are available.

Weak instruments pose a further source of bias. A genetic variant is considered a weak instrument if it has a weak statistical association with the exposure relative to the sample size, resulting in finite-sample bias in the causal effect estimate [12]. The magnitude and direction of this bias depends on the degree of sample overlap between the exposure and outcome GWASs. In a complete overlap (one-sample) setting, causal estimates are biased toward the observational association, which may increase false positive findings. In contrast, using two non-overlapping samples biases results toward the null. While most summary-level MR methods have been designed under the assumption of complete independence between exposure and outcome GWAS samples, substantial participant overlap is common in practice as the largest outcome and exposure GWAS often come from the same consortia [13]. Restricting analyses to non-overlapping GWASs again leads to inefficient data use.

To address these challenges, we propose a novel summary-level MR method, MR Simulated Sample Splitting (MR-SimSS). MR-SimSS extends the benefits of the 'three-sample' design to single sample settings, by imitating the process of splitting an individual-level dataset into three parts: one for instrument selection and two for independent estimation of the variant-exposure and variant-outcome associations. Importantly, this is achieved using only GWAS summary statistics. MR-SimSS leverages asymptotic conditional distributions to repetitively simulate association estimates in each of the three parts, or data-subsets, conditional on the known full dataset estimates. On each iteration, instrument variants are selected based on the simulated variant-exposure associations for the first subsample, and a two-sample MR method, such as IVW or MR-RAPS, is then applied to the exposure and outcome associations that are simulated for that set of instruments in the second and third subsets. MR estimates from each iteration are averaged to reduce variance. This approach allows use of the full dataset while avoiding biases introduced by sample overlap and Winner's Curse.

We assessed MR-SimSS under varying levels of sample overlap and exposure-outcome correlation, using pairs of simulated exposure and outcome GWAS summary statistics. Several existing MR methods were included for comparison. Same-trait analyses were also conducted using UK Biobank [14] body mass index (BMI) data, where the true causal effect is known to be 1, providing a useful benchmark. All methods were evaluated across three key metrics: average bias, root mean square error (RMSE) and empirical coverage probability of 95% confidence intervals. These evaluations showed that many MR methods are adversely affected by biases arising due to sample overlap, be it using the same samples for instrument selection and estimating variant-exposure associations or using overlapping datasets to estimate both variant-exposure and variant-outcome associations. MR-SimSS illustrated a strong ability to mitigate these biases, especially those arising from Winner's Curse. However, the simulated sample splitting process can increase susceptibility to weak instrument bias. To address this, we recommend incorporating a robust two-sample method, such as MR-RAPS [10], within MR-SimSS. Our results show that the 3-split version of MR-SimSS, when combined with MR-RAPS, consistently outperforms all other evaluated methods across key performance metrics. Finally, we demonstrate the practical utility of MR-SimSS in a real-world example assessing the causal effect of BMI on risk of Type 2 diabetes (T2D), highlighting the method's value in applied epidemiological research.

## Methods

### Overview of MR-SimSS

MR-SimSS reconstructs the statistical conditions of a three-sample Mendelian Randomization (MR) design using only GWAS summary statistics for variant-exposure ($\hat{\beta}_X$) and variant-outcome ($\hat{\beta}_Y$) associations, even when these originate

from overlapping samples. This enables unbiased causal inference in the presence of Winner's Curse and weak instrument bias. The method operates under the assumption that marginal variant effect estimates follow an asymptotic multivariate normal distribution. Based on this, we have derived an analytical expression (see Equation (1)) that allows simulation of marginally independent variant effect estimates from hypothetical non-overlapping subsamples, without requiring access to individual-level genotypes or phenotypes.

The core procedure comprises three steps. First, MR-SimSS defines a hypothetical partition of the original dataset, allocating a fraction (e.g., $\pi_1$) to a synthetic discovery subset. Conditional on the observed summary statistics, variant-exposure and variant-outcome association estimates ($\hat{\beta}_X^1$, $\hat{\beta}_Y^1$) are simulated for this subset and used to select instruments based on a genome-wide significance threshold.

Second, to mitigate Winner's Curse, MR-SimSS generates variant-exposure and variant-outcome association estimates ($\hat{\beta}_X^2$, $\hat{\beta}_Y^2$) from the remaining pseudo-subsample. These are derived according to the asymptotic linearity of maximum likelihood estimators under data partitioning (see Equation (2)), and are marginally independent of the selection step. As a result, they can be supplied directly to any standard two-sample MR estimator, such as IVW or MR-RAPS, without inducing selection bias.

However, in the presence of sample overlap between exposure and outcome GWASs, a residual correlation between $\hat{\beta}_X^2$ and $\hat{\beta}_Y^2$ may remain, reintroducing weak instrument bias in the direction of the confounded observational association. To address this, MR-SimSS incorporates a three-split extension, where the $1 - \pi_1$ data fraction is further subdivided into two non-overlapping fractions. Variant-exposure association estimates are generated for one fraction, while independent variant-outcome association estimates are generated for the other fraction, using a similar conditional Gaussian framework, this time conditioning on the simulated summary statistics $\hat{\beta}_X^2$ and $\hat{\beta}_Y^2$ from the first step (see Equation (S24)). This removes residual covariance and restores the assumption of independence between numerator and denominator that is made by many two-sample MR estimators, like IVW and MR-RAPS. This step is especially important when exposure and outcome GWASs exhibit substantial sample overlap, ensuring that any remaining bias, if not corrected by the chosen MR method, is toward the null.

To improve stability, the entire procedure is repeated over multiple random simulated splits, and causal effect estimates are averaged across iterations. MR-SimSS is compatible with both continuous and binary traits and generalizes to a wide range of GWAS settings, including full or partial sample overlap. Importantly, it permits valid use of robust MR methods, such as MR-RAPS, in settings where standard assumptions of independence are violated. Unbiased and efficient estimation of causal effects using commonly available summary-level data is therefore enabled as MR-SimSS reconstructs the independence structure of a three-sample MR design via conditional simulation.

## Technical details

For each genetic variant $j = 1, \ldots, N,$ we assume availability of variant-exposure and variant-outcome association estimates together with their respective standard errors, i.e., $\left\{ \hat{\beta}_{X_j}, \hat{\sigma}_{X_j} \right\}$ from an exposure GWAS with sample size $n_X$ and $\left\{ \hat{\beta}_{Y_j}, \hat{\sigma}_{Y_j} \right\}$ from an outcome GWAS with sample size $n_Y$. Summary statistics are assumed to arise from linear or logistic regression models applied to standardized genotypes, following standard GWAS practice. We allow for possible sample overlap ($n_{\text{overlap}} \leq \min(n_X, n_Y)$) between the exposure and outcome studies, and assume that linkage disequilibrium (LD) pruning has already been applied, resulting in a set of uncorrelated variants with summary statistics available in both GWASs.

If this individual-level data were available, Winner's Curse could be eliminated by randomly splitting the data into two fractions $\pi_1$ and $1 - \pi_1$, conducting GWASs on each, and then selecting instruments in one part and estimating causal effects in the other. Since only summary-level data are available, we simulate this splitting process by drawing from our

derived asymptotic conditional distribution of the GWAS estimates in the $\pi_1$-fraction, conditional on the full-sample GWAS statistics. Therefore, for each variant $j$, association estimates $\hat{\beta}_{X_j}^1$ and $\hat{\beta}_{Y_j}^1$ are simulated according to:

$$\begin{pmatrix}\hat{\beta}_{X_j}^1 \\ \hat{\beta}_{Y_j}^1\end{pmatrix} \Bigg| \begin{pmatrix}\hat{\beta}_{X_j} \\ \hat{\beta}_{Y_j}\end{pmatrix} \sim N\left(\begin{pmatrix}\hat{\beta}_{X_j} \\ \hat{\beta}_{Y_j}\end{pmatrix}, \frac{1-\pi_1}{\pi_1}\begin{pmatrix}\sigma_{X_j}^2 & \lambda\sigma_{X_j}\sigma_{Y_j} \\ \lambda\sigma_{X_j}\sigma_{Y_j} & \sigma_{Y_j}^2\end{pmatrix}\right) \tag{1}$$

where $\lambda = \frac{n_{\text{overlap}}\rho}{\sqrt{n_X n_Y}}$ and $\rho = \text{cor}(X, Y)$ denotes the correlation between the exposure and outcome, potentially non-zero due to confounding. A proof that this is the correct conditional distribution to use for the simulation is given in S1 Text. In practice, $n_{\text{overlap}}$ and $\rho$ may not be known, and therefore, we propose a data-driven estimator of the parameter $\lambda = \frac{n_{\text{overlap}}\rho}{\sqrt{n_X n_Y}}$, as detailed in S1 Text. Unconditional standard errors in the $\pi_1$-fraction are approximated by $\sigma_{X_j}^1 \approx \frac{\hat{\sigma}_{X_j}}{\sqrt{\pi_1}}$ and $\sigma_{Y_j}^1 \approx \frac{\hat{\sigma}_{Y_j}}{\sqrt{\pi_1}}$. Instruments are selected using z-statistics $z_{X_j}^1 = \frac{\hat{\beta}_{X_j}^1}{\sigma_{X_j}^1}$, and a genome-wide significance threshold of $5 \times 10^{-8}$.

To ensure independence between instrument selection and estimation, association estimates in the remaining $(1-\pi_1)$-fraction can be reconstructed as follows, using asymptotic linearity of maximum likelihood estimates:

$$\hat{\beta}_{X_j}^2 \approx \frac{\hat{\beta}_{X_j} - \pi_1\hat{\beta}_{X_j}^1}{1-\pi_1}, \quad \hat{\beta}_{Y_j}^2 \approx \frac{\hat{\beta}_{Y_j} - \pi_1\hat{\beta}_{Y_j}^1}{1-\pi_1} \tag{2}$$

with corresponding standard errors scaled by $\frac{1}{\sqrt{1-\pi_1}}$. In the simplest (2-split) implementation of MR-SimSS, the selected genetic variants and their corresponding association estimates in the $(1-\pi_1)$-fraction are inputted into a summary-level MR method, such as IVW. However, this approach may still incur weak instrument bias in the direction of confounding if the exposure and outcome GWASs overlap.

To address this, we extend the approach to a 3-split framework. In this version, the $(1-\pi_1)$-fraction is further conceptually split into sub-fractions of relative sizes $\pi_2$ and $1-\pi_2$, with simulated exposure estimates, $\hat{\beta}_{X_j}^{2a}$, derived from the $\pi_2$-subset and outcome estimates, $\hat{\beta}_{Y_j}^{2b}$, from the remaining $(1-\pi_2)$-subset. These are simulated using a second conditional distribution, analogous to the form above (see Equation (S24) in S1 Text). For each variant $j$, this yields independent association estimates and associated standard errors, $\left\{\hat{\beta}_{X_j}^{2a}, \sigma_{X_j}^{2a}, \hat{\beta}_{Y_j}^{2b}, \sigma_{Y_j}^{2b}\right\}$, which are then used as input into the 2-sample MR method being used with MR-SimSS at each iteration. Again, standard errors are scaled appropriately with $\sigma_{X_j}^{2a} \approx \frac{\hat{\sigma}_{X_j}}{\sqrt{\pi_2(1-\pi_1)}}$ and $\sigma_{Y_j}^{2b} \approx \frac{\hat{\sigma}_{Y_j}}{\sqrt{(1-\pi_2)(1-\pi_1)}}$. This simulated sample splitting procedure is repeated multiple times to reduce variability, and the final causal effect estimate, $\hat{\beta}$, is computed by averaging across iterations $k = 1, \ldots, N_{\text{iter}}$:

$$\hat{\beta} = \frac{1}{N_{\text{iter}}} \sum_{k=1}^{N_{\text{iter}}} \hat{\beta}^{(k)} \tag{3}$$

in which $\hat{\beta}^{(k)}$ is the causal effect estimate supplied by the summary-level MR method of choice on the $k^{\text{th}}$ iteration. To quantify uncertainty, we derive the standard error of the average estimate using the following decomposition:

$$\text{se}(\hat{\beta}) = \sqrt{\frac{1}{N_{\text{iter}}} \sum_{k=1}^{N_{\text{iter}}} [\text{se}(\hat{\beta}^{(k)})]^2 - \frac{1}{N_{\text{iter}}} \sum_{k=1}^{N_{\text{iter}}} [\hat{\beta}^{(k)} - \hat{\beta}]^2} \tag{4}$$

The first term captures the average estimation variance across iterations, while the second adjusts for between-iteration variation, analogous to a Monte Carlo error correction. This formulation follows from the identity $\text{Var}(X) = \mathbb{E}\left[X^2\right] - (\mathbb{E}[X])^2$, and is shown in more detail in S1 Text. We refer to this method as MR-SimSS (Mendelian Randomization via Simulated Sample Splitting). The default implementation sets $N_{\text{iter}} = 1000$ to ensure stable convergence of the mean and variance

estimates, and fixes $\pi_1 = \pi_2 = 0.5$, following empirical guidance from Sadreev et al. [15], who found equal splits to offer the most flexibility when designing two-sample MR with sample splitting.

Because the 3-split procedure may increase weak instrument bias towards the null due to reduced effective sample sizes, we also consider MR-SimSS in combination with MR-RAPS [10], which is designed to account for weak instruments when using independent samples. Our framework facilitates the application of MR-RAPS even when the underlying GWASs are partially overlapping, by producing independent summary statistics through simulation.

While MR-SimSS is designed to improve causal inference accuracy, it can be computationally demanding when applied to large GWAS datasets due to the need to repeatedly simulate and evaluate many variants. To mitigate this, we introduce a deterministic variant pre-selection strategy that retains computational efficiency while preserving instrument coverage with high probability. For each variant $j$, we compute the probability that it will be selected in any iteration based on the simulated z-statistic $z_j^1 = \frac{\hat{\beta}_{X_j}^1}{\sigma_{X_j}^1}$. The probability of variant $j$ passing the significance threshold $\Phi^{-1}(1 - \alpha/2)$, for the standard normal cumulative distribution function $\Phi(\cdot)$ and chosen $\alpha$, is given by:

$$P\left(\left|Z_j^1\right| > \Phi^{-1}\left(1 - \frac{\alpha}{2}\right) \mid z_j\right) = \Phi\left(\frac{-\Phi^{-1}\left(1 - \frac{\alpha}{2}\right) + z_j}{\sqrt{1 - \pi_1}}\right) + \Phi\left(\frac{-\Phi^{-1}\left(1 - \frac{\alpha}{2}\right) - z_j}{\sqrt{1 - \pi_1}}\right)$$

(5)

in which $z_j = \frac{\hat{\beta}_{X_j}}{\sqrt{\frac{1}{\pi_1}} \cdot \sigma_{X_j}}$. To construct a reduced variant subset, we rank variants by their selection probabilities and compute the cumulative sum of these probabilities. We retain the smallest set of variants whose cumulative inclusion probability exceeds 0.95. This guarantees that the reduced and full variant sets will produce identical instruments on any iteration with at least 95% probability. Equivalently, the expected difference in the number of instruments selected by the full and reduced procedures is bounded by 5%. This subsetting procedure provides a principled and efficient means to scale MR-SimSS to large-scale genomic subsets. Note that a more complete description of the strategy is available in S1 Text.

## Simulation study

We conducted simulations to assess the performance of MR-SimSS in reducing Winner's Curse bias in MR and to compare it against established MR methods. A factorial design was implemented, varying the following parameters:

- Exposure heritability: $h^2 \in \{0.3, 0.7\}$

- Proportion of causal SNPs: $p \in \{0.01, 0.001\}$

- Sample overlap: $n_{overlap} \in \{0, 0.25, 0.5, 0.75, 1\} \cdot n_X$

- Exposure-outcome correlation: $\rho \in \{-0.1, 0.1, 0.3, 0.5\}$

Each scenario assumed equal exposure and outcome GWAS sample sizes, $n_X = n_Y = 200,000$, and a true causal effect $\beta = 0.3$. For each replicate, we simulated GWAS summary statistics for $N = 1,000,000$ independent genetic variants. True variant-exposure effects $\beta_{X_j}$ were sampled such that a proportion $p$ had non-zero effects drawn from a normal distribution, calibrated to yield the specified heritability $h^2$. True variant-outcome effects were defined as $\beta_{Y_j} = \beta \cdot \beta_{X_j}$.

Estimated summary statistics $\left(\hat{\beta}_{X_j}, \hat{\beta}_{Y_j}\right)$ were drawn from a bivariate normal distribution:

$$\begin{pmatrix} \hat{\beta}_{X_j} \\ \hat{\beta}_{Y_j} \end{pmatrix} \sim N\left(\begin{pmatrix} \beta_{X_j} \\ \beta_{Y_j} \end{pmatrix}, \frac{1}{2maf_j(1 - maf_j)}\begin{pmatrix} \frac{1}{n_X} & \frac{n_{overlap}\rho}{n_X n_Y} \\ \frac{n_{overlap}\rho}{n_X n_Y} & \frac{1}{n_Y} \end{pmatrix}\right)$$

(6)

with variant minor allele frequencies ($\mathsf{maf}_j$) simulated uniformly over [0.01, 0.5], an expression that is justified as an appropriate asymptotic distribution in S1 Text, when both exposure and outcome have variance 1. Standard errors were assumed to be known. For each of the 80 parameter combinations, we simulated 100 independent datasets. To assess robustness to sample size, the simulations were repeated for $n_X = n_Y = \{50,000, \ 500,000\}$. In addition, we conducted simulations under the null hypothesis ($\beta = 0$) with complete sample overlap, as well as simulations examining different pleiotropic scenarios.

Each dataset was analysed using MR-SimSS (2-split and 3-split) with IVW and MR-RAPS, as well as standard MR methods such as IVW and MR-RAPS using genome-wide significant instruments ($p < 5 \times 10^{-8}$). Each of these methods was evaluated using:

- Bias = $\frac{1}{100} \sum_{k=1}^{100} (\hat{\beta}^{(k)} - \beta)$

- Root Mean Squared Error (RMSE) = $\sqrt{\frac{1}{100} \sum_{k=1}^{100} (\hat{\beta}^{(k)} - \beta)^2}$

- Coverage = $\frac{1}{100} \sum_{k=1}^{100} \mathbb{I}\{(\hat{\beta}^{(k)} - 1.96 \cdot \mathsf{se}(\hat{\beta}^{(k)})) < \beta < (\hat{\beta}^{(k)} + 1.96 \cdot \mathsf{se}(\hat{\beta}^{(k)}))\}$

Here, $\beta = 0.3$ is the true causal effect, $\hat{\beta}^{(k)}$ and $\mathsf{se}(\hat{\beta}^{(k)})$ are the estimate and standard error in replicate $k$, and $\mathbb{I}$ is the indicator function. We additionally report the average causal effect estimate, the average standard error and the average absolute bias (absolute estimated bias averaged over simulation settings).

### Real data processing

For the empirical same-trait BMI-BMI analyses, the large-scale UK Biobank [14] BMI dataset was randomly split in half 10 times to generate 10 pairs of non-overlapping samples. In each of the 20 resulting subsets, quality control and GWAS were performed using PLINK 2.0 [16], following the same procedures as outlined in Forde et al. [17]. This yielded 10 pairs of independent GWAS summary statistics datasets. A set of approximately independent variants was obtained via LD pruning using the PLINK 2.0 [16] command 'indep-pairwise 50 5 0.5'. Instrument variants were selected based on the conventional genome-wide significance threshold of $p < 5 \times 10^{-8}$.

## Results

### Simulation study

We conducted a comprehensive simulation study to evaluate the finite-sample performance of MR-SimSS relative to existing summary-level MR methods. The simulations assumed equal exposure and outcome GWAS sample sizes of 200,000, and explored 80 distinct configurations defined by varying the proportion of causal variants, heritability of the exposure, fraction of sample overlap, and exposure-outcome correlation. Throughout, it was assumed that overlap and correlation parameters were unknown to the analyst. For each simulation, MR-SimSS in both two-split and three-split variants using MR methods; IVW and MR-RAPS, was used to estimate the causal effect of the exposure on the outcome. The implementation of MR-SimSS also incorporated estimation of the parameter λ, representing the correlation between variant-exposure and variant-outcome association estimates (see Methods).

Table 1 summarizes mean performance metrics - bias, absolute bias, root mean squared error (RMSE) and 95% coverage probability - averaged over heritability and proportion of true effects, for non-overlapping and fully overlapping samples with an exposure-outcome correlation of 0.5. Figs 1 and 2 present boxplots of estimated causal effects for the non-overlapping and fully overlapping scenarios, respectively, stratified by exposure-outcome correlation, heritability and proportion of true effects. Under non-zero sample overlap, SimSS-3-RAPS, the three-split implementation incorporating MR-RAPS, consistently exhibited superior performance. In simulations with complete sample overlap and an exposure-outcome correlation of 0.5, SimSS-3-RAPS achieved minimal bias (-0.0001), the lowest RMSE (0.0068), and the highest empirical coverage (93.5%). When exposure and outcome samples were independent, SimSS-2-RAPS

**Table 1. Summarized simulation results for each method for non-overlapping and fully overlapping samples with exposure-outcome correlation = 0.5, averaged over heritability and proportion of true effect variants.**

| Method | $\hat{\beta}$ | bias | \|bias\| | RMSE | SE | CP | # IVs |
|---|---|---|---|---|---|---|---|
| *Zero Overlap* | | | | | | | |
| **SimSS-2-IVW** | 0.2935 | -0.0065 | 0.0076 | 0.0105 | 0.0062 | 0.8000 | 367.41 |
| **SimSS-2-RAPS** | 0.2997 | -0.0003 | 0.0051 | 0.0076 | 0.0063 | 0.9600 | 367.39 |
| **SimSS-3-IVW** | 0.2875 | -0.0125 | 0.0126 | 0.0161 | 0.0063 | 0.4550 | 367.39 |
| **SimSS-3-RAPS** | 0.2997 | 0.0003 | 0.0053 | 0.0078 | 0.0063 | 0.9400 | 367.39 |
| **IVW** | 0.2849 | -0.0151 | 0.0154 | 0.0209 | 0.0045 | 0.4350 | 764.37 |
| **RAPS** | 0.2886 | -0.0114 | 0.0122 | 0.0171 | 0.0046 | 0.5025 | 764.37 |
| *Full Overlap* | | | | | | | |
| **SimSS-2-IVW** | 0.3040 | 0.0040 | 0.0056 | 0.0079 | 0.0068 | 0.9225 | 367.32 |
| **SimSS-2-RAPS** | 0.3087 | 0.0087 | 0.0092 | 0.0119 | 0.0077 | 0.7925 | 367.29 |
| **SimSS-3-IVW** | 0.2878 | -0.0123 | 0.0124 | 0.0155 | 0.0056 | 0.4025 | 367.31 |
| **SimSS-3-RAPS** | 0.2999 | -0.0001 | 0.0045 | 0.0068 | 0.0055 | 0.9350 | 368.03 |
| **IVW** | 0.3101 | 0.0101 | 0.0105 | 0.0142 | 0.0043 | 0.4825 | 764.91 |
| **RAPS** | 0.3130 | 0.0130 | 0.0133 | 0.0174 | 0.0046 | 0.4350 | 764.91 |

Methods are abbreviated as: SimSS-2-IVW = 2-split version of MR-SimSS using IVW, SimSS-2-RAPS = 2-split version of MR-SimSS using MR-RAPS, SimSS-3-IVW = 3-split version of MR-SimSS using IVW, SimSS-3-RAPS = 3-split version of MR-SimSS using MR-RAPS, IVW = Inverse variance weighted method and RAPS = Robust Adjusted Profile Score of Zhao et al. [10]. The columns are: average estimated effect size ($\hat{\beta}$), average bias (bias), average absolute bias (|bias|), root mean squared error (RMSE), average standard error (SE), empirical coverage probability of 95% confidence intervals (CP), and average number of instruments used (# IVs). Light green shaded cells highlight the method that provided the best result for each evaluation metric.

outperformed competing methods, attaining the lowest average bias and RMSE (-0.0003 and 0.0076, respectively) and high empirical coverage (96%; Table 1). These performance patterns were consistent across all values of the exposure-outcome correlation (Figs 1-2).

Although MR-SimSS corrects for Winner's Curse, the simulated sample splitting procedure reduces the effective sample size used for estimation, increasing susceptibility to weak instrument bias when paired with IVW. This effect was evident for SimSS-3-IVW, which uses simulated non-overlapping splits for the generation of variant-exposure and variant-outcome association estimates. This MR-SimSS variant exhibited consistent downward bias regardless of sample overlap, with causal estimates remaining below 0.3 in all simulation settings (Figs 1-2). In contrast, SimSS-2-IVW shows overlap-dependent bias, reflecting incomplete independence between the simulated subsamples used for estimation.

Incorporating MR-RAPS within MR-SimSS mitigates this limitation. Under independent or weakly overlapping samples, both SimSS-2-RAPS and SimSS-3-RAPS yielded nearly unbiased estimates (Fig 1, Fig B in S1 Text), consistent with the theoretical properties of MR-RAPS under independence of variant-exposure and variant-outcome associations [10]. However, under high sample overlap and exposure-outcome correlation (0.5), SimSS-2-RAPS exhibited inflated bias (Fig 2), whereas SimSS-3-RAPS remained unbiased due to the enforced independence of simulated exposure and outcome associations used in the estimation step. In the extreme case of complete sample overlap and strong exposure-outcome correlation, SimSS-3-RAPS was the only method to avoid upward bias toward the observational association, aside from the negatively biased SimSS-3-IVW. Across all MR-SimSS variants, standard errors were modestly larger than those of competing methods, reflecting variance inflation from simulated splitting. In contrast, the standard IVW and MR-RAPS estimators exhibit substantial bias across multiple simulation settings, reflecting susceptibility to both Winner's Curse and bias arising from sample overlap (Figs 1-2).

**Zero Overlap**

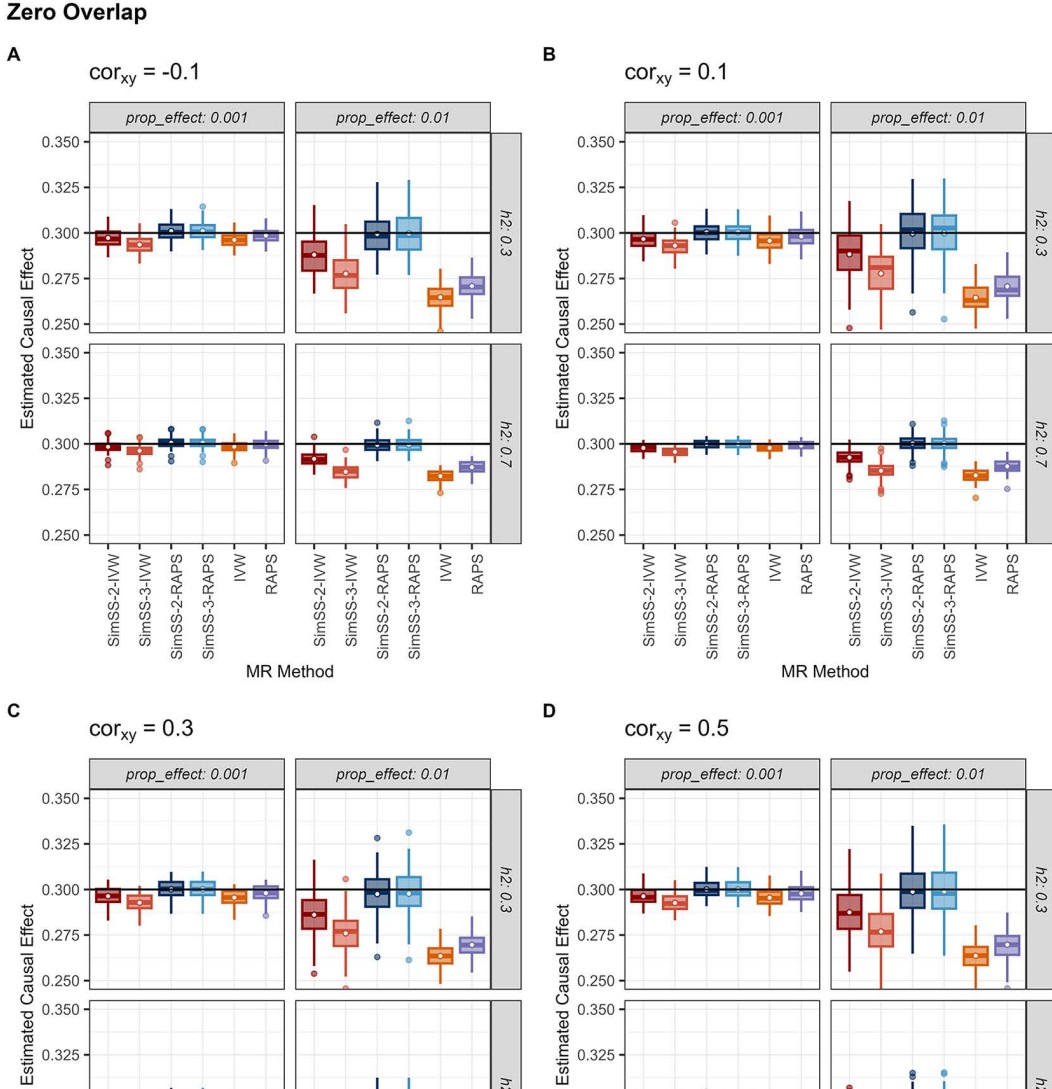

**Fig 1. Causal effect estimates for simulation settings with zero overlap.** Estimated causal effect for each method and simulation setting with sample sizes of 200,000 and zero overlap, averaged over 100 simulated pairs of exposure and outcome GWAS summary statistics for each setting. Methods are abbreviated as: SimSS-2-IVW = 2-split version of MR-SimSS using IVW, SimSS-2-RAPS = 2-split version of MR-SimSS using MR-RAPS, SimSS-3-IVW = 3-split version of MR-SimSS using IVW, SimSS-3-RAPS = 3-split version of MR-SimSS using MR-RAPS, IVW = Inverse variance weighted method and RAPS = Robust Adjusted Profile Score of Zhao et al. [10].

**Full Overlap**

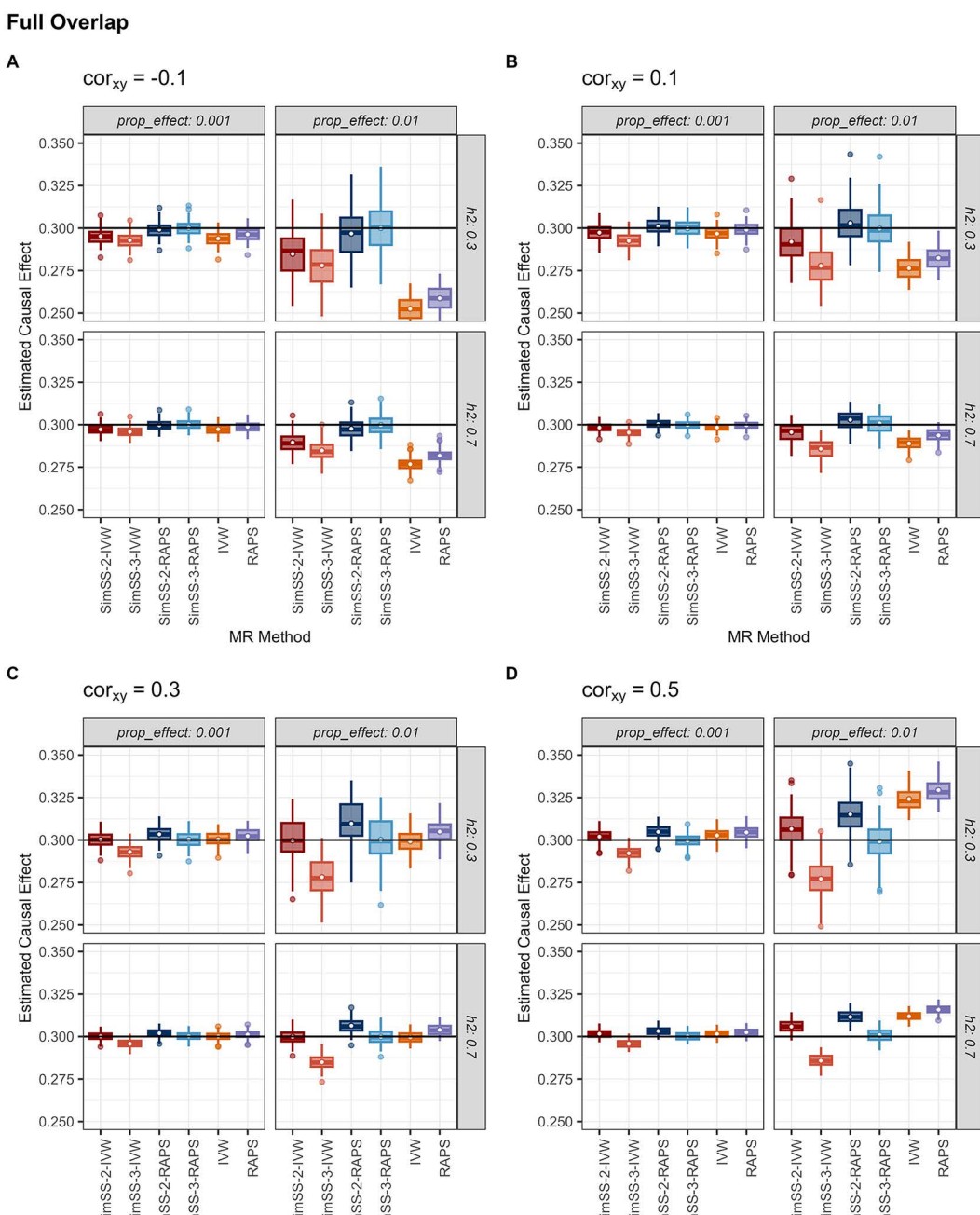

**Fig 2. Causal effect estimates for simulation settings with full overlap.** Estimated causal effect for each method and simulation setting with sample sizes of 200,000 and full overlap, averaged over 100 simulated pairs of exposure and outcome GWAS summary statistics for each setting. Methods are abbreviated as: SimSS-2-IVW = 2-split version of MR-SimSS using IVW, SimSS-2-RAPS = 2-split version of MR-SimSS using MR-RAPS, SimSS-3-IVW = 3-split version of MR-SimSS using IVW, SimSS-3-RAPS = 3-split version of MR-SimSS using MR-RAPS, IVW = Inverse variance weighted method and RAPS = Robust Adjusted Profile Score of Zhao et al. [10].

Fig 1 and Table B in S1 Text report performance under strictly non-overlapping samples for all other exposure-outcome correlation values. In these settings, both SimSS-2-RAPS and SimSS-3-RAPS achieved near-zero bias (<0.0005), low RMSE, and empirical coverage between 94% and 97%. SimSS-2-RAPS achieved slightly lower RMSE due to the absence of unnecessary variance inflation from an additional split. Conversely, standard IVW and MR-RAPS were substantially biased in scenarios with low proportions of causal variants. In the opposite extreme of complete sample overlap (Fig 2, Table C in S1 Text), SimSS-3-RAPS again achieved the most favourable balance of bias, RMSE, and coverage, with respect to all values of exposure-outcome correlation. Performance of all methods under intermediate overlap fractions (25%, 50%, 75%) is shown in Figs B-D and Tables D-F in S1 Text, in which consistent trends favouring SimSS-3-RAPS are evident.

Fig E and Table G in S1 Text report results from simulations conducted under the null hypothesis of no causal effect with fully overlapping samples. These results highlight the impact of biases due to instrument selection and participant overlap when two-sample MR methods are naively implemented, with IVW and MR-RAPS exhibiting poor empirical coverage and markedly elevated false positive rates. In contrast, both SimSS-3-IVW and SimSS-3-RAPS achieved approximately 95% coverage, indicating preservation of Type I error at the nominal 5% level under this setting.

To assess sensitivity to sample size, method performance was evaluated under alternative sample sizes of 500,000 and 50,000. With larger samples (500,000), performance improved across all methods due to increased instrument strength. Nevertheless, SimSS-3-RAPS remained the most accurate estimator, achieving minimal average absolute bias (0.0021), an RMSE of 0.0029, and empirical coverage of 94.1%. In contrast, for smaller sample sizes of 50,000, MR-SimSS methods exhibited sensitivity to instrument sparsity. Although SimSS-3-RAPS remained the least biased estimator (average absolute bias = 0.0012), it exhibited substantially increased variability, with a standard error of approximately 58.29 and an RMSE more than tenfold those of IVW and MR-RAPS (Table F in S1 Text). This instability is further illustrated in Fig H in S1 Text, where causal effect estimates span a wide range (-1–1), reflecting the limited number of genome-wide significant variants in these settings. For example, under 30% heritability and a 1% true effect proportion, only approximately 10 variants exceeded genome-wide significance, yielding an average of around three instruments per iteration, and thus, severely impairing causal effect estimation with MR-SimSS.

To investigate whether the performance of SimSS-3-RAPS could be improved under such low-power conditions, we examined the effect of relaxing the instrument selection threshold. As shown in Fig K and Table J in S1 Text, increasing the significance threshold from $5 \times 10^{-8}$ to $5 \times 10^{-4}$ substantially reduced RMSE, from 0.923 to 0.032, while maintaining low average absolute bias (0.0258). These findings indicate that relaxing the selection threshold within MR-SimSS can improve estimator stability when few variants reach genome-wide significance in the first sample split ($p < 5 \times 10^{-8}$). However, because such adjustments introduce weak instrument bias, it is essential that robust MR methods, such as MR-RAPS, are used within the MR-SimSS framework to ensure valid inference. Note that additional results from simulations examining various forms of pleiotropy are provided in Tables K-P and Figs L-O in S1 Text. These results further demonstrate MR-SimSS' ability to avoid Winner's Curse and overlap-induced bias, while retaining the properties of the embedded two-sample MR method.

### Same-trait empirical analysis

To assess the empirical performance of MR-SimSS, we conducted same-trait MR analyses, estimating the causal effect of BMI on itself, using independent GWASs from the UK Biobank [14]. These analyses provide a realistic validation setting in which the true causal effect is known to be 1. 10 pairs of non-overlapping samples of ~166,000 individuals were used to generate BMI GWAS summary statistics for ~1.6 million LD-pruned variants using PLINK 2.0 [16]. Same-trait MR analyses were performed bidirectionally for each pair of summary statistics, yielding 20 causal effect estimates per evaluated method. All MR-SimSS variants and conventional methods, including MR-Egger and weighted median [6], were assessed using bias, RMSE and 95% coverage probability. Additional methods evaluated here also included debiased IVW (dIVW),

a bias-corrected estimator that uses all variants and does not rely on SNP selection [18], and MRlap, a likelihood-based method designed to correct for both Winner's Curse and sample overlap bias [19].

Table 2 and Fig 3 summarize the results of the repeated BMI-BMI analyses using pruned instruments. Classical IVW, MR-RAPS and weighted median estimators exhibited substantial downward bias, with mean causal estimates of 0.838, 0.865, and 0.807, respectively. All three methods yielded 0% empirical coverage, confirming susceptibility to Winner's Curse. MR-Egger was less biased with an average causal effect estimate of 0.989, but its coverage was moderate (0.65)

**Table 2. Summarized results for the sets of 20 same-trait BMI-BMI analyses for each method.**

| Method | $\hat{\beta}$ | bias | \|bias\| | RMSE | SE | CP | # IVs |
|---|---|---|---|---|---|---|---|
| SimSS-2-IVW | 0.9683 | -0.0317 | 0.0317 | 0.0362 | 0.0153 | 0.50 | 93.05 |
| SimSS-3-IVW | 0.9392 | -0.0608 | 0.0608 | 0.0632 | 0.0156 | 0.00 | 92.97 |
| SimSS-2-RAPS | 0.9928 | -0.0073 | 0.0144 | 0.0189 | 0.0177 | 0.95 | 93.01 |
| SimSS-3-RAPS | 0.9931 | -0.0070 | 0.0148 | 0.0194 | 0.0153 | 0.85 | 92.94 |
| IVW | 0.8381 | -0.1619 | 0.1619 | 0.1636 | 0.0107 | 0.00 | 470.75 |
| Egger | 0.9899 | -0.0101 | 0.0607 | 0.0683 | 0.0332 | 0.65 | 470.75 |
| Weighted median | 0.8072 | -0.1928 | 0.1928 | 0.1960 | 0.0143 | 0.00 | 470.75 |
| RAPS | 0.8649 | -0.1351 | 0.1351 | 0.1370 | 0.0091 | 0.00 | 470.75 |
| dIVW | 0.9974 | -0.0026 | 0.0124 | 0.0146 | 0.0042 | 0.25 | 1,589,444 |
| MRlap | 1.0277 | 0.0277 | 0.0302 | 0.0361 | 0.0338 | 0.90 | 89.25 |

Methods are abbreviated as: SimSS-2-IVW = 2-split version of MR-SimSS using IVW, SimSS-2-RAPS = 2-split version of MR-SimSS using MR-RAPS, SimSS-3-IVW = 3-split version of MR-SimSS using IVW, SimSS-3-RAPS = 3-split version of MR-SimSS using MR-RAPS, IVW = Inverse variance weighted method, RAPS = Robust Adjusted Profile Score of Zhao et al. [10], Egger = Egger regression of Bowden et al. [9], Weighted median = Weighted median approach of Bowden et al. [6], dIVW = debiased IVW method of Ye et al. [18] and MRlap = MRlap method of Mounier & Kutalik [19]. The columns are: average estimated effect size ($\hat{\beta}$), average bias (bias), average absolute bias (|bias|), root mean squared error (RMSE), average standard error (SE), empirical coverage probability of 95% confidence intervals (CP), and average number of instruments used (# IVs).

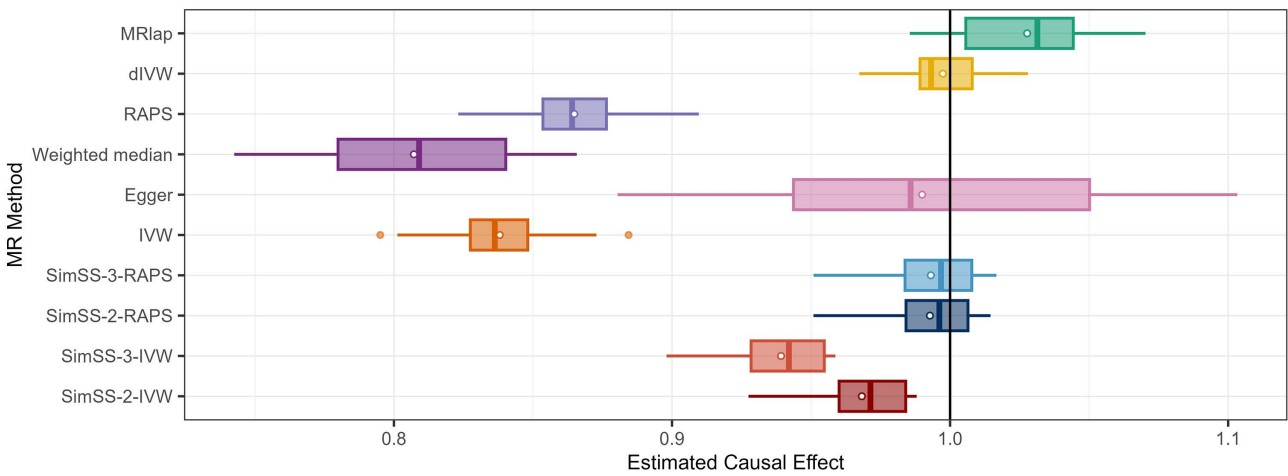

**Fig 3. Causal effect estimates for 20 same-trait BMI-BMI analyses.** Boxplots of the estimated causal effects for each method resulting from the 20 same-trait BMI-BMI analyses. Methods are abbreviated as: SimSS-2-IVW = 2-split version of MR-SimSS using IVW, SimSS-2-RAPS = 2-split version of MR-SimSS using MR-RAPS, SimSS-3-IVW = 3-split version of MR-SimSS using IVW, SimSS-3-RAPS = 3-split version of MR-SimSS using MR-RAPS, IVW = Inverse variance weighted method, RAPS = Robust Adjusted Profile Score of Zhao et al. [10], Egger = Egger regression of Bowden et al. [9], Weighted median = Weighted median approach of Bowden et al. [6], dIVW = debiased IVW method of Ye et al. [18] and MRlap = MRlap method of Mounier & Kutalik [19]. The black horizontal line represents the true causal effect of 1.

and it exhibited large estimate variance, reflecting instability. MRlap also produced estimates close to the true causal effect (1.028), with high empirical coverage (0.9), although it showed upward bias and increased variability, suggesting potential violation of model assumptions, e.g., spike-and-slab genetic architecture assumption. While dIVW achieved the lowest bias (-0.0026) and RMSE, its poor coverage (0.25) indicates miscalibrated standard errors.

In contrast, SimSS-2-RAPS and SimSS-3-RAPS provided a favourable balance between bias, precision and calibration. Both approaches yielded near-unbiased estimates (bias = -0.0073 and -0.0070, respectively), with substantially lower RMSE than classical estimators and high empirical coverage (85–95%). SimSS-2-IVW and SimSS-3-IVW eliminated Winner's Curse-induced attenuation but suffered from downward weak instrument bias, with mean biases of -0.0247 and -0.0503, and reduced coverage. Overall, these empirical results reinforce simulation findings: MR-SimSS, particularly when implemented with MR-RAPS, can offer a robust correction for selection-induced bias, with the ability to deliver accurate and well-calibrated causal estimates even in real-world data.

## Effect of body mass index on type 2 diabetes

The four MR-SimSS variants, together with other MR methods, were also used to estimate the causal effect of BMI on type 2 diabetes (T2D), under varying degrees of sample overlap. First, two independent samples of ~166,000 individuals with outcome information were randomly selected from the entire UKBB [14] T2D data set and used to generate two sets of outcome GWAS summary statistics (T2D-A and T2D-B) with PLINK 2.0 [16]. For the exposure, BMI, 5 different sets of GWAS summary statistics were generated using similarly-sized sets of individuals, all with different percentages of sample overlap with the outcome data sets (0%, 25%, 50%, 75%, 100%).

For each overlap setting, results of the corresponding BMI-T2D analyses were pooled to provide average estimated causal effects. These results are summarized in Fig 4 and Table 3. In line with previous MR studies [20], higher BMI was confirmed to be a causal risk factor for T2D. All methods yield statistically significant causal effect estimates, with all

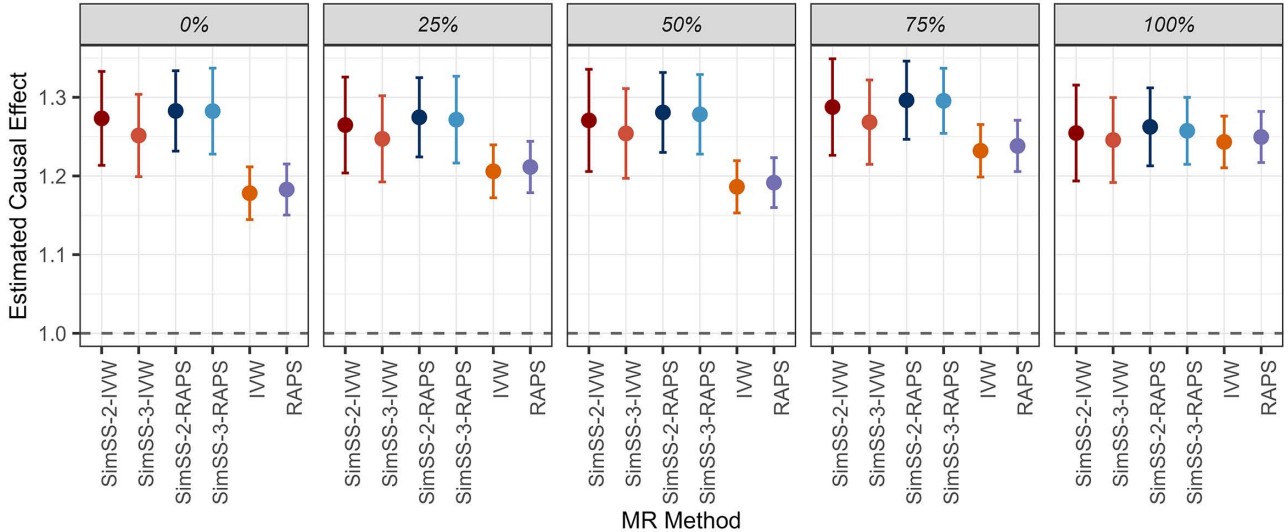

**Fig 4. Causal effect estimates for BMI-T2D analyses, with varying sample overlap.** Average estimated causal effects for each method resulting from BMI-T2D analyses with various percentages of sample overlap. Error bars reflect confidence intervals computed as: $\hat{\beta} \pm 1.96 \cdot (\text{average SE})$. Methods are abbreviated as: SimSS-2-IVW = 2-split version of MR-SimSS using IVW, SimSS-2-RAPS = 2-split version of MR-SimSS using MR-RAPS, SimSS-3-IVW = 3-split version of MR-SimSS using IVW, SimSS-3-RAPS = 3-split version of MR-SimSS using MR-RAPS, IVW = Inverse variance weighted method, and RAPS = Robust Adjusted Profile Score of Zhao et al. [10].

**Table 3. Summarized results for the BMI-T2D analyses across varying sample overlap.**

| % Overlap | 0 | 25 | 50 | 75 | 100 | |
|---|---|---|---|---|---|---|
| | $\hat{\beta}$ (SE) | $\hat{\beta}$ (SE) | $\hat{\beta}$ (SE) | $\hat{\beta}$ (SE) | $\hat{\beta}$ (SE) | # IVs |
| SimSS-2-IVW | 1.2732 (0.0304) | 1.2647 (0.0311) | 1.2706 (0.0331) | 1.2876 (0.0313) | 1.2546 (0.0311) | 93.27 |
| SimSS-3-IVW | 1.2515 (0.0267) | 1.2472 (0.0279) | 1.2541 (0.0291) | 1.2684 (0.0274) | 1.2457 (0.0276) | 93.18 |
| SimSS-2-RAPS | 1.2827 (0.0261) | 1.2746 (0.0257) | 1.2807 (0.0259) | 1.2963 (0.0254) | 1.2624 (0.0253) | 93.24 |
| SimSS-3-RAPS | 1.2824 (0.0279) | 1.2716 (0.0281) | 1.2784 (0.0258) | 1.2955 (0.0211) | 1.2574 (0.0217) | 93.35 |
| IVW | 1.1781 (0.0171) | 1.2059 (0.0172) | 1.1862 (0.0170) | 1.2320 (0.0170) | 1.2432 (0.0168) | 472.4 |
| RAPS | 1.1828 (0.0165) | 1.2114 (0.0167) | 1.1915 (0.0162) | 1.2382 (0.0166) | 1.2495 (0.0166) | 472.4 |
| dIVW | 1.2673 (0.0062) | | | | | 1,589,444 |

Average estimated BMI-T2D causal effects with average standard errors (italics) generated by each method for five different percentages of sample overlap.

95% confidence intervals lying above 1. The traditional IVW approach yielded estimated effects ranging from 1.178, for non-overlapping samples, to 1.243, for fully overlapping samples. In contrast, the range of causal effect estimates provided by SimSS-3-RAPS is∼40% smaller, from 1.257 to 1.296. The standard deviation of SimSS-3-RAPS averaged estimates (0.014) was less than half of that of IVW (0.0282), giving evidence that SimSS-3-RAPS can provide more consistent effect estimates across different degrees of sample overlap between exposure and outcome data sets. The difference between SimSS-3-RAPS and IVW estimates was greatest in the zero overlap setting, with our results suggesting that IVW underestimated the causal effect by ~8% due to downward bias caused by both Winner's Curse and weak instruments.

## Discussion

We introduce MR Simulated Sample Splitting (MR-SimSS), a novel summary-level MR method designed to correct Winner's Curse bias. Winner's Curse arises when the same GWAS dataset is used for both instrument selection and variant-exposure association estimation, often producing deflated causal effect estimates. In recent years, summary-level MR has become the dominant MR approach, driven by its ease of use and the broad availability of complete summary data. This has spurred the development of numerous summary-level MR methods aimed at accurate causal effect estimation [21]. However, methods explicitly addressing Winner's Curse bias remain underdeveloped. While using an independent sample for instrument selection is a commonly accepted solution [10], we consider such an approach suboptimal due to reduced statistical power from dataset partitioning and potential heterogeneity introduced by dissimilar populations. Accordingly, MR-SimSS requires no such independent dataset to be available.

MR-SimSS addresses Winner's Curse by employing asymptotic conditional distributions to emulate repetitive splitting of a large individual-level dataset into distinct fractions. Instrument selection is based on association estimates from one fraction, while the remaining fraction is used for estimation. At each iteration, causal effect estimates are obtained via the integration of a summary-level MR method. We investigated the use of both the IVW [7] and MR-RAPS [10] methods within the context of our simulated sample splitting procedure. Furthermore, the MR-SimSS framework accommodates both partial and full sample overlap between exposure and outcome GWASs, a frequent scenario in large biobank

datasets such as UK Biobank. We note that similar approaches to controlling for sample overlap have been previously proposed within the GRAPPLE framework [22].

In a factorial simulation study, varying sample overlap and exposure-outcome correlation, SimSS-3-RAPS - the 3-split version of MR-SimSS integrating MR-RAPS - consistently demonstrated superior performance, yielding unbiased causal estimates across all scenarios. It achieved the highest empirical coverage, minimal average bias and lowest RMSE, illustrating its capacity to overcome both Winner's Curse and weak instrument bias. As shown in Fig 2, SimSS-3-RAPS remains unbiased irrespective of exposure-outcome correlation, even in the presence of fully overlapping samples. When independent GWASs are available, SimSS-2-RAPS performs comparably well. Although SimSS-2-IVW and SimSS-3-IVW are susceptible to weak instrument bias, both outperform naïve IVW, underscoring MR-SimSS's utility even when paired with simpler estimators. Our simulations reaffirm the vulnerability of standard methods, such as IVW and MR-RAPS, to Winner's Curse. For example, Table 1 and Table B in S1 Text shows coverage below 0.51 for these methods under zero-overlap conditions when selection and estimation utilize the same exposure data.

Our simulation results with smaller sample sizes, e.g., 50,000, underscore the importance of sufficient instrument strength and quantity for stable estimation using MR-SimSS. When the average number of instruments per iteration is low, fewer than ~20, the method exhibits increased variability and reduced precision. However, we find that performance improves substantially when the instrument selection threshold is relaxed. Specifically, our findings demonstrate that in low-power settings, employing a less stringent selection threshold offers a practical strategy to recover estimator stability within the MR-SimSS framework (Fig K in S1 Text), provided that weak instrument bias is appropriately mitigated through the use of robust MR estimators. However, this logic only extends so far. For instance, we would not recommend applying MR-SimSS blindly if there are no genome-wide exposure-associated variants. MR-SimSS with a reduced selection threshold is likely to have extremely low power in this setting, even under large exposure-outcome causal effects, and possibly inflated type I error.

Our simulations also demonstrate that MR-SimSS in its three-split variety, when paired with MR-RAPS has roughly 95% coverage, and consequentially 5% Type I error, under the null hypothesis of no causal effect (Fig E in S1 Text). While it is true that in zero overlap scenarios, classical MR methods, such as IVW, can demonstrate superior power over MR-SimSS variants when the true causal effect is small, these methods are highly susceptible to bias when sample overlap exists, resulting in greatly inflated Type I errors (Figs P-Q in S1 Text). This Type I error inflation is also seen for methods that are robust to weak instrument bias under no sample overlap, such as MR-RAPS, as we show in our simulations. In contrast, our study shows that SimSS-3-RAPS has the ability to retain a 5% Type I error, irrespective of the degree of sample overlap. Given what has been referred to as a 'credibility crisis' in Mendelian Randomization [23], it is essential that MR methods have preserved Type I error under general conditions, and thus, the MR-SimSS framework is an important development in this regard.

Same-trait empirical analyses (e.g., BMI-BMI) corroborate the simulation findings, particularly under independent sample conditions (Fig 3). Across 20 BMI-BMI analyses, IVW, MR-RAPS, and the weighted median method displayed pronounced Winner's Curse bias, with coverage probabilities of zero and average bias ranging from -0.2 to -0.13. In contrast, SimSS-2-RAPS and SimSS-3-RAPS yielded unbiased estimates with high coverage and negligible bias (<0.01), demonstrating robustness to both Winner's Curse and weak instrument bias.

In BMI-T2D analyses, MR-SimSS produced more stable and consistent causal effect estimates across varying degrees of sample overlap than conventional approaches. In particular, SimSS-3-RAPS yielded a narrower range of estimates with lower variability than standard IVW, suggesting improved robustness to both Winner's Curse and weak instrument bias. For comparative purposes, the debiased IVW (dIVW) estimator [18] was applied in the zero-overlap setting. The dIVW method explicitly corrects for measurement error in variant-exposure associations and doesn't rely on screening variants, thereby avoiding selection bias due to Winner's Curse. Therefore, dIVW estimates obtained under sample independence should theoretically be similar to those provided by SimSS-3-RAPS in all overlap settings. The resulting averaged

dIVW estimate lay within the range of estimates produced by SimSS-3-RAPS (Table 3), providing further support that MR-SimSS effectively mitigates both selection- and overlap-induced bias when applied to real data. We note that this empirical study has focussed primarily on alleviating bias arising from Winner's Curse and sample overlap and thus, the potential impact of other sources of bias, such as correlated pleiotropy, remains an avenue for further exploration.

Admittedly, the current formulation of MR-SimSS has certain limitations. Throughout our investigation, the splitting fractions, $\pi_1$ and $\pi_2$, were both fixed at 0.5. Determining universally optimal values for these fractions is inherently challenging as they likely depend on the underlying genetic architectures of both the exposure and outcome traits, as well as the sample sizes of the source GWAS datasets. Our selection of $\pi_1 = 0.5$ was partly informed by Sadreev et al. [15]. A higher value of $\pi_1$ increases the number of instruments per iteration; thus, $\pi_1 = 0.5$ balances the fractions used for instrument generation and for estimation of associations in the 3-split setting. With both splitting fractions set to 0.5, only 25% of the total sample informs each variant-exposure and variant-outcome association estimate, increasing variance in the resulting causal effect estimates. This variance inflation can be largely mitigated by using a sufficiently large number of iterations, e.g., 1000, ensuring stability and precision in the final estimate. Adaptive tuning of these parameters merits future exploration.

An additional practical consideration when applying MR-SimSS concerns the handling of linkage disequilibrium (LD) among genetic variants. In its current form, we recommend implementing MR-SimSS with an LD-pruned, rather than LD-clumped, set of variants, as pruning avoids additional selection bias introduced by retaining only the most strongly exposure-associated variant within each LD region and ensures that the standard measurement-error model for the selected variant-exposure associations remains appropriate. To illustrate this, we performed a same-trait BMI-BMI analysis using both pruned and clumped variant sets with comparable numbers of genome-wide significant variants (Fig R, Table Q in S1 Text). The MR-SimSS approaches incorporating MR-RAPS yielded unbiased causal effect estimates when applied to pruned variants, but not when applied to clumped variants. This observation is consistent with the additional selection bias induced by clumping, which preferentially selects variants with the smallest p-values within each genomic region and thus, biases estimates toward the null. In principle, this selection bias could be mitigated by explicitly modelling correlation between variant-exposure associations, for example by injecting correlated noise according to the underlying LD structure. Exploring such extensions, which would allow MR-SimSS to completely remove Winner's Curse bias from clumped datasets, forms an interesting direction for future methodological work.

Another avenue for future research is the integration of MR-SimSS with alternative summary-level MR methods. Because MR-SimSS inherits the properties of the embedded two-sample MR method, optimal performance is expected when it is paired with methods that are robust to weak instruments and pleiotropy, whereas more vulnerable methods may yield less reliable estimates. This is illustrated by SimSS-3-RAPS, whose strong resistance to weak instrument bias is derived directly from MR-RAPS. Accordingly, combining MR-SimSS with pleiotropy-robust methods is expected to be appropriate for analyses affected by directional or horizontal pleiotropy. To explore this, we conducted an auxiliary simulation study incorporating pleiotropy, in which MR-SimSS was combined with the MR weighted median estimator [6] (SimSS-2-Med and SimSS-3-Med). Although this implementation did clearly exhibit resistance to pleiotropy-induced bias, its estimates remained biased downwards due to the weighted median's susceptibility to weak instrument bias (Figs N-O, Tables M-P in S1 Text). Conceptually, MR-SimSS should therefore be viewed not as a standalone MR estimator, but as a general framework for mitigating bias due to Winner's Curse and sample overlap in summary-level MR. It complements existing MR methods, enabling researchers to utilize the largest available GWAS datasets, even with overlapping samples, without requiring a third, independent dataset.

Due to the lack of accessible software, we were unable to include the recently proposed rerandomized IVW (RIVW) estimator [24] in our method evaluations. The RIVW estimator may be regarded as conceptually similar to MR-SimSS, as both approaches were designed to facilitate independent instrument selection and unbiased estimation of variant-exposure associations using a single exposure dataset. By introducing pseudo variant-exposure associations into the

selection step and then using Rao-Blackwellization to produce a consistent estimator for the causal effect, RIVW successfully breaks the Winner's Curse in the classical two-sample IVW estimator [24]. For settings with non-overlapping exposure and outcome samples, this Rao-Blackwellization result could be viewed as the theoretical expectation of what would be obtained if MR-SimSS was applied with an IVW estimate adapted to handle weak instrument bias. As RIVW is a non-simulation based approach, it is likely to be more computationally efficient than MR-SimSS. However, it requires that the exposure and outcome GWASs have been performed with non-overlapping samples and is vulnerable to unbalanced pleiotropy. In contrast, we have demonstrated here how MR-SimSS can mitigate Winner's Curse bias irrespective of sample overlap and can also be used with existing MR methods that are resistant to weak instrument bias and certain types of pleiotropy.

In conclusion, MR-SimSS provides a principled and practical solution to two pervasive issues in MR analysis, Winner's Curse and weak instrument bias, when only GWAS summary statistics are accessible. By enabling the use of maximal GWAS data, MR-SimSS substantially improves the reliability of causal effect estimates, thus offering a certain valuable contribution to the MR methodological toolkit.

## Supporting information

**S1 Text. Text supplement.** This file contains supplementary figures, tables as well as important derivations. (PDF)

## Author contributions

**Conceptualization:** Gibran Hemani, John Ferguson.

**Data curation:** Amanda Forde.

**Formal analysis:** Amanda Forde.

**Investigation:** Amanda Forde.

**Methodology:** Amanda Forde, John Ferguson.

**Resources:** John Ferguson.

**Software:** Amanda Forde.

**Supervision:** Gibran Hemani, John Ferguson.

**Visualization:** Amanda Forde.

**Writing – original draft:** Amanda Forde.

**Writing – review & editing:** Amanda Forde, Gibran Hemani, John Ferguson.

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
