## [Decision Letter · Decision Letter 0]

1 Dec 2025

PGENETICS-D-25-01198

Simulated sample splitting approach to address biases due to instrument selection and participant overlap in two-sample Mendelian Randomization studies

PLOS Genetics

Dear Dr. Forde,

Thank you for submitting your manuscript to PLOS Genetics. After careful consideration, we feel that it has merit but does not fully meet PLOS Genetics's publication criteria as it currently stands. Therefore, we invite you to submit a revised version of the manuscript that addresses the points raised during the review process.

We look forward to receiving your revised manuscript.

Kind regards,

Xiang Zhou, Ph.D.

Academic Editor

PLOS Genetics

Michael Epstein

Section Editor

PLOS Genetics

Aimée Dudley

Editor-in-Chief

PLOS Genetics

Anne Goriely

Editor-in-Chief

PLOS Genetics

**Journal Requirements:**

At this stage, the following Authors/Authors require contributions: Amanda Forde, Gibran Hemani, and John Ferguson. Please ensure that the full contributions of each author are acknowledged in the "Add/Edit/Remove Authors" section of our submission form.

The list of CRediT author contributions may be found here: https://journals.plos.org/plosgenetics/s/authorship#loc-author-contributions

3) Kindly revise your competing statement in the online submission form to align with the journal's style guidelines: 'The authors declare that there are no competing interests.'

**Reviewers' comments:**

Reviewer's Responses to Questions

**Comments to the Authors:**

Reviewer #1: This is a creative and timely solution to the winner's curse problem in MR. I enjoyed reading the paper, and have only minor comments.

1. Table 1 basically shows no bias in any method except dIVW, which isn't very useful. I suppose it's because it averages over situations with different degrees of overlap. It may be more informative to break this table down a bit, will full results still in supplement.

2. The dIVW results are not really comparable with the others because it is using the full set of 1M SNPs. I appreciate that it needs a large number of SNPs to get a good bias adjustment, but this might explain some of the differences from other methods. MR-RAPS can also work with a large number of SNPs, so if you want to consider whole-genome instruments, you could perhaps compare dIVW to MR-RAPS (others have found them to be very similar).

3. Supp p17, "By standard linear model theory.." - var(X) should be the residual variance (which is approx var(X) when SNPs have small effects). Also, the normal distribution is asymptotic.

4. Supp p18, below eq S9 the sentence beginning "Hence, " is not rigorous - I know what you mean, but it's illogical to define C as approximately equal to something.

5. Supp p19, eq S19 the 4th expression should be var instead of cov? (can't have covariance of one variable)

6. Supp p20, above eq S24, "For binary Y..." there is no error term in the logistic regression equation

7. Supp p21 first line, is the W on the RHS correct?

Reviewer #2: See attachment.

Reviewer #3: The paper presents an original and smart idea how to simulate summary statistics from sample splitting and uses this to reduce Winner’s curse bias in Mendelian Randomisation. I find the m/s well-written, the idea truly original and the results are sound. While it does not seem to be a game-changer for well-powered MR studies, it has numerous merits and deserves publication. Below, I list a few comments that could help improving the paper.

Major comments (in arbitrary order):

Since the authors already simulate a 3way split of the samples, can’t they use the first split to select the SNPs and from the second split they could use the fraction “p” for the exposure and fraction”1-p” for the outcome and do the reverse as well? Would this ensure that there is no sample overlap?

The simulation exercise seems to be too simple, there is no pleiotropy introduced, while in real settings both correlated and uncorrelated pleiotropy is frequent. These need to be explored.

Another relevant methods, such as MRlap [https://pubmed.ncbi.nlm.nih.gov/37036286/] would be interesting to test against MR-SimSS, since it claims to handle sample overlap and winner’s curse related biases.

The authors should investigate the power of these methods. Given the settings, IVW would not produce false positives, thus its discovery power is key to be compared to that of MR-SimSS. Simulations show a very small downward bias of IVW (~3% relative bias), but 25% lower SE compared to MR-SimSS, which implies superior power for IVW.

Since the main disadvantage of MR-SimRR is the sample split, hence lower power, it should be emphasised more that the method is not recommended for MR with N below 100k (or less than 20 IVs)?

It should be acknowledged in the Discussion that the approach cannot help random over-/under-estimation of SNP effects that is present in the full sample (before splitting) and by definition their method will still preferentially select SNPs whose effects were overestimated in the full sample.

While in the particular simulation setting using milder IV selection thresholds does not seem to have a detrimental impact on the causal effect estimation, in general it is rather dangerous to do, since it depends on the genetic architecture how many null SNPs are selected by mistake during this step, which dilutes the causal effect (or biases it in arbitrary direction in case of sample overlap). Thus, in general, this strategy should not be encouraged and readers must be warned of this in the Discussion.

The BMI->T2D results are not Discussed - I’d be curious to hear what conclusions can be drawn from those differences about the true causal effect and specific weaknesses of the methods.

Minor comment:

It is a bit confusing the \pi is used both to denote polygenicity, and also to refer to the sample split proportion. I’d introduce another variable for one of them.

**Have all data underlying the figures and results presented in the manuscript been provided?**

Reviewer #1: Yes

Reviewer #2: None

Reviewer #3: Yes

PLOS authors have the option to publish the peer review history of their article (what does this mean?). If published, this will include your full peer review and any attached files.

Reviewer #1: No

Reviewer #2: **Yes:** Qingyuan Zhao

Reviewer #3: No

**Figure resubmission:**
---

## [Decision Letter · Decision Letter 1]

24 Mar 2026

PGENETICS-D-25-01198R1

Simulated sample splitting approach to address biases due to instrument selection and participant overlap in two-sample Mendelian Randomization studies

PLOS Genetics

Dear Dr. Forde,

Thank you for submitting your manuscript to PLOS Genetics. After careful consideration, we feel that it has merit but does not fully meet PLOS Genetics's publication criteria as it currently stands. Therefore, we invite you to submit a revised version of the manuscript that addresses the minor points raised during the review process.

Please submit your revised manuscript within by Apr 23 2026 11:59PM. If you will need more time than this to complete your revisions, please reply to this message or contact the journal office at plosgenetics@plos.org. Please include the following items when submitting your revised manuscript:

We look forward to receiving your revised manuscript.

Kind regards,

Xiang Zhou, Ph.D.

Academic Editor

PLOS Genetics

Michael Epstein

Section Editor

PLOS Genetics

Aimée Dudley

Editor-in-Chief

PLOS Genetics

Anne Goriely

Editor-in-Chief

PLOS Genetics

**Reviewers' comments:**

Reviewer's Responses to Questions

**Comments to the Authors:**

Reviewer #1: Thank you for addressing my comments.

One possible error: on line 217 of the clean version, I don't think the square root sign should extend over sigma.

Reviewer #2: I have two remaining comments:

1. It's an interesting proposal to use LD pruning to address my second comment. I still think LD clumping after injecting correlated noise is likely a better approach, but understand the computational concern. But I'm worried about the power of LD pruning --- in my experience if one sets a stringent LD threshold, LD pruning tends to miss many signals; if one sets a loose LD threshold, LD pruning tends to produce correlated instruments and methods assuming independent instruments tend to underestimate the standard error. Do the authors observe something similar in their BMI-BMI analysis (or simulation analyses)?

2. I hope the authors will consider publishing the response letter which in my opinion contains useful information for users and methodologists.

Reviewer #3: I congratulate the authors on their thorough revision, greatly addressing my points.

In case the decision is minor revision, I have a small comment, but it is not crucial to address:

“The fact that there maybe more null-effect variants selected as instruments should not cause

bias for the same reason, provided some genuine instruments are selected.” - I would thus include a warning that applying this (or any other MR approach blindly) even if there are no genome-wide significant instruments are present is not recommended.

**Have all data underlying the figures and results presented in the manuscript been provided?**

Reviewer #1: None

Reviewer #2: None

Reviewer #3: Yes

PLOS authors have the option to publish the peer review history of their article (what does this mean?). If published, this will include your full peer review and any attached files.

Reviewer #1: No

Reviewer #2: **Yes:** Qingyuan Zhao

Reviewer #3: **Yes:** Zoltan Kutalik

**Figure resubmission:**
---

## [Decision Letter · Decision Letter 2]

26 Apr 2026

Dear Dr Forde,

We are pleased to inform you that your manuscript entitled "Simulated sample splitting approach to address biases due to instrument selection and participant overlap in two-sample Mendelian Randomization studies" has been editorially accepted for publication in PLOS Genetics. Congratulations!

Yours sincerely,

Xiang Zhou, Ph.D.

Academic Editor

PLOS Genetics

Michael Epstein

Section Editor

PLOS Genetics

Aimée Dudley

Editor-in-Chief

PLOS Genetics

Anne Goriely

Editor-in-Chief

PLOS Genetics

BlueSky: @plos.bsky.social

Comments from the reviewers (if applicable):

Reviewer's Responses to Questions

**Comments to the Authors:**

Reviewer #2: Nothing further, congratulations on an excellent contribution.

**Have all data underlying the figures and results presented in the manuscript been provided?**

Reviewer #2: None

PLOS authors have the option to publish the peer review history of their article (what does this mean?). If published, this will include your full peer review and any attached files.

Reviewer #2: **Yes:** Qingyuan Zhao

**Data Deposition**

http://datadryad.org/submit?journalID=pgenetics&manu=PGENETICS-D-25-01198R2

**Press Queries**

---

## [Editor Report · Acceptance letter]

PGENETICS-D-25-01198R2

Simulated sample splitting approach to address biases due to instrument selection and participant overlap in two-sample Mendelian Randomization studies

Dear Dr Forde,

We are pleased to inform you that your manuscript entitled "Simulated sample splitting approach to address biases due to instrument selection and participant overlap in two-sample Mendelian Randomization studies" has been formally accepted for publication in PLOS Genetics! Your manuscript is now with our production department and you will be notified of the publication date in due course.

With kind regards,

Zsofia Freund

PLOS Genetics

On behalf of:
